# OpenReview forum: "AgentAlign: Navigating Safety Alignment in the Shift from Informative to Agentic Large Language Models"
_ICLR.cc/2026/Conference — Submitted to ICLR 2026_

### Official Review · Reviewer_fXJL · 2025-10-20

**Soundness:** 3
**Presentation:** 3
**Contribution:** 3
**Rating:** 4
**Confidence:** 3

**Summary:**

This work introduces AgentAlign, a new training framework to improve agent safety in which the attackers instruct the agents to execute malicious tasks. They show positive results where AgentAlign outperformed basic prompting defense baselines such as CoT, ReAct, and refusal prompts.

**Strengths:**

- Solid synthetic data generation pipeline with human validations.
- Great diagrams and plots that help explain things clearly.
- Showed great performance when compared to prompting baselines.

**Weaknesses:**

- Only compared to weak prompting baselines. For example, I'd appreciate it if you could at least add one common guardrail baseline, such as Llama Guard 4? This is a common defense method and will be helpful to include to see whether it complements AgentAlign or it's already very effective enough on the benchmarks evaluated.
- No adversarial pressure studied. In reality, attackers will not give up after one try, and will likely apply the existing, common jailbreaking approach to the original harmful instructions. It'd be great to see a section that studies how robust this training approach is to for example, automated red teaming, multi-turn or decomposition attack, prefill attack, etc.
- There is no limitation section. This work studies the single-turn attack for the agents, but it's unclear whether this works for multi-turn attack, etc. some papers you can cite in multi-turn or decomposition attacks include: (1) Monitoring Decomposition Attacks in LLMs with
Lightweight Sequential Monitors and (2) Breach By A Thousand Leaks: Unsafe Information Leakage in `Safe' AI Responses
- No error bars were provided in any of the stats reported.

**Questions:**

Questions are basically the weakness i mentioned:
- Could you include llama guard 4 as a baseline?
- Could you test this approach on a few common jailbreaking techniques mentioned in Weakness 2?
- Could you add a limitation section?
- Could you at least report the standard errors for table 2? because the baseline and AgentAlign are quite close.

Happy to raise my score if all of these are addressed, and AgentAlign still outperforms the new baseline and still holds the positive results under the adversarial pressure.

---

> ### Author Response · Authors · 2025-11-22
> **Response to Weakness**
>
> Thank you for your careful assessment of our work and the feedback you've provided. We appreciate your thoughtful review. Below are our responses to the questions you raised, which we hope will address your concerns.
>
> > Only compared to weak prompting baselines. For example, I'd appreciate it if you could at least add one common guardrail baseline, such as Llama Guard 4? This is a common defense method and will be helpful to include to see whether it complements AgentAlign or it's already very effective enough on the benchmarks evaluated.
>
> - Following your suggestion, we have added comparative results with LlamaGuard. To ensure a fair comparison, we benchmarked our AgentAlign-trained Functionary-Small-v3.2 (based on Llama-3-8B) against Llama-Guard-3-8B:
>
>   |                        | Harmful Requests Refusal (%) | Benign Requests Refusal (%) |
>   | :--------------------: | :--------------------------: | :-------------------------: |
>   |    Llama-Guard-3-8B    |             97.7             |            24.4             |
>   | Functionary-8B-aligned |             88.6             |            12.5             |
>
>   As the results demonstrate, Llama-Guard-3-8B suffers from significant overfitting to harmful content, leading to a high false positive rate of 24% on benign user requests—a critical limitation that severely impacts practical usability. Our method, by contrast, successfully internalizes safety awareness while maintaining superior utility, achieving a more favorable safety-utility balance. Furthermore, **our approach enables direct deployment without the architectural overhead of an additional external guardrail module, providing a more efficient and self-contained solution**.
>
> >  No adversarial pressure studied. In reality, attackers will not give up after one try, and will likely apply the existing, common jailbreaking approach to the original harmful instructions. It'd be great to see a section that studies how robust this training approach is to for example, automated red teaming, multi-turn or decomposition attack, prefill attack, etc.
>
> - We appreciate this important observation. In our pilot studies, we tested the aligned models against adversarial jailbreak methods such as **[Adaptive Jailbreaking](https://openreview.net/forum?id=hXA8wqRdyV)**. Under these attacks, our models' robustness degraded to approximately the base model level (or marginally better). We attribute this to the fact that jailbreak defense is highly case-specific and typically requires dedicated adversarial training—an orthogonal research direction to our core focus. Our work primarily addresses the foundational safety alignment gap that arises when transitioning from conversational to agentic LLMs, which we tackle through novel mechanisms including abstract behavior chains and  training based on rollouts in simulated environments. **We believe that without first establishing this fundamental alignment layer, addressing advanced attacks is premature if basic safety alignment remains unsolved.** That said, we agree this is an important limitation and will add an honest discussion of adversarial robustness to the Limitations section in our revision. Thank you for this valuable feedback.
>
> > There is no limitation section. This work studies the single-turn attack for the agents, but it's unclear whether this works for multi-turn attack, etc. some papers you can cite in multi-turn or decomposition attacks include: (1) Monitoring Decomposition Attacks in LLMs with Lightweight Sequential Monitors and (2) Breach By A Thousand Leaks: Unsafe Information Leakage in `Safe' AI Responses
>
> - Thank you for this this suggestion. We will include a new Limitations section discussing adversarial robustness as well as multi-turn and decomposition attack approaches, providing a more complete picture of our method's current scope and open challenges.
>
> > No error bars were provided in any of the stats reported.
>
> - We have added error bars to the experiments in the response to questions to better illustrate the statistical variance and significance of our results.

---

> ### Author Response · Authors · 2025-11-22
> **Response to Questions**
>
> > Could you include llama guard 4 as a baseline?
>
> - We have included a comparison with the same-sized Llama-Guard-3-8B model for a fair evaluation.
>
> > Could you test this approach on a few common jailbreaking techniques mentioned in Weakness 2?
>
> - Please see our our response to weakness 2, where we elaborate on this concern.
>
> > Could you add a limitation section?
>
> - Yes, we will include a Limitations section as mentioned above, covering adversarial robustness and the multi-turn/decomposition attack scenarios you highlighted.
>
> > Could you at least report the standard errors for table 2? because the baseline and AgentAlign are quite close.
>
> - We have added standard error analyses to Table 2 (Evaluation on ToolSword) as you suggested. We ran 5 independent experiments for the Qwen-2.5-7B-Instruct vs. AgentAlign comparison:
>
>   |    Toolsword Eval    | Mean (%) | SE (%) |  n   |
>   | :------------------: | :------: | :----: | :--: |
>   | Qwen-2.5-7b-Instruct |  96.36   |  0.58  |  5   |
>   |     +AgentAlgin      |  98.91   |  0.43  |  5   |
>
>   The results show that our safety-aligned model achieves not only higher safety scores but also better stability (lower standard errors). We note that the relatively close performance on ToolSword can be attributed to its origin as an adaptation of existing red-teaming datasets. On the more challenging AgentHarm benchmark, which better captures agentic safety complexities, our AgentAlign models show substantially stronger performance, better demonstrating the value of our approach.

---

> ### Comment · Reviewer_fXJL · 2025-11-23
>
> Thank you for the detailed responses.
>
> > Following your suggestion, we have added comparative results with LlamaGuard.
>
> Thank you for running this experiment! Given these numbers, it is not clear that AgentAlign actually achieves a better safety–utility trade-off than llama guard 4. While AgentAlign has a lower false-positive rate, it also has a substantially weaker harmful-request refusal rate (≈ 89% vs. 98%), which is a major part of safety. Given this, I'm not convinced the proposed training approach is notably more valuable than the existing safety method.
>
> >   In our pilot studies, we tested the aligned models against adversarial jailbreak methods such as Adaptive Jailbreaking. Under these attacks, our models' robustness degraded to approximately the base model level (or marginally better). We attribute this to the fact that jailbreak defense is highly case-specific and typically requires dedicated adversarial training—an orthogonal research direction to our core focus.
>
> Thank you for the response; however, I want to note that adaptive jailbreak is not tied to a specific jailbreaking technique; rather, it's a general strategy that iteratively adjusts the prompt to achieve jailbreaking. So I'm not sure if attributing the vulnerability of the proposed method when applying adaptive attack to "jailbreak defense is highly case-specific," is sound.
> Lastly, how is adaptive attack orthogonal to your research direction? Isn't this work fundamentally trying to improve the model so that it will refuse any adversarial prompts in the agentic setup? And that if the proposed method is not robust enough under the known strong adversarial pressure (adaptive attack https://arxiv.org/abs/2510.09023 ), I, as a reader, am not sure if this contributes to the safety community in a significant way.
>
> > Thank you for this this suggestion. We will include a new Limitations section discussing adversarial robustness as well as multi-turn and decomposition attack approaches, providing a more complete picture of our method's current scope and open challenges.
>
> Could you please just add this Limitations section in the rebuttal version?
>
> > We have added standard error analyses to Table 2 (Evaluation on ToolSword) as you suggested. We ran 5 independent experiments for the Qwen-2.5-7B-Instruct vs. AgentAlign comparison:
>
> I appreciate this.
>
> Overall, given the concerns outlined above, I'm not comfortable raising my score.

---

> > ### Author Response · Authors · 2025-11-28
> >
> > > Given these numbers, it is not clear that AgentAlign actually achieves a better safety–utility trade-off than llama guard 4.
> >
> > - We have included additional evaluation results using Llama-Guard-4 and found that its performance is even less satisfactory (incorrectly rejecting 43% of benign requests). We hypothesize that this degradation may be attributed to the introduction of multimodal capabilities.
> >
> >   |                        | Harmful Requests Refusal (%) | Benign Requests Refusal (%) |
> >   | :--------------------: | :--------------------------: | :-------------------------: |
> >   |    Llama-Guard-3-8B    |             97.7             |            24.4             |
> >   |   Llama-Guard-4-12B    |             93.8             |            43.2             |
> >   | Functionary-8B-aligned |             88.6             |            12.5             |
> >
> > > While AgentAlign has a lower false-positive rate, it also has a substantially weaker harmful-request refusal rate (≈ 89% vs. 98%), which is a major part of safety. Given this, I'm not convinced the proposed training approach is notably more valuable than the existing safety method.
> >
> > - In fact, during our early experiments fine-tuning Llama-3 with AgentAlign, we observed similar results to Llama-Guard-3: a very high refusal rate on harmful instructions but also an extremely high false positive rate on benign tasks. However, we believe this trade-off is impractical—a 9% improvement came at the cost of a 12% increase in false positives, which would severely impact user experience. Therefore, we adjusted the data composition to achieve a more practical balance. For example, **Functionary-AgentAlign achieves both a higher true refusal rate (88.6% vs. 86.4%) and a lower false positive rate (12.5% vs. 15.9%) compared to Claude-3.5-Haiku**.
> >
> > - Beyond achieving a superior safety-utility balance, our work offers the following advantages over guardrail-based methods:
> >
> >   - **It eliminates the need for additional deployment and inference overhead, making the approach more efficient and self-contained**.
> >   - **It serves as a valuable resource even for enhancing agentic capabilities**. As shown in Table 1, our method substantially enhances performance on benign tasks for both Qwen-2.5-7B (53.4% → 64.2%) and Functionary-Small-v3.2/Llama-3.1-8B (45.9% → 53.5%).
> >
> > > I'm not sure if attributing the vulnerability of the proposed method when applying adaptive attack to "jailbreak defense is highly case-specific," is sound. And that if the proposed method is not robust enough under the known strong adversarial pressure (adaptive attack https://arxiv.org/abs/2510.09023 ), I, as a reader, am not sure if this contributes to the safety community in a significant way.
> >
> > - Our statement that "jailbreak defense is highly case-specific" is based on the following considerations: Jailbreaking techniques like Adaptive Jailbreaking are likely amenable to mitigation through approaches such as  [Instruction Hierarchy](https://arxiv.org/pdf/2404.13208) or other adversarial training methods, whereas attacks like GCG can be detected through perplexity checks or LLM-based detection.
> >
> >   However, **what we want to emphasize is the reality that current LLMs, when serving as agents, respond to harmful instructions without requiring any jailbreaking techniques.** This means that a much broader population of ordinary users with no knowledge of jailbreaking can readily exploit alignment gaps in these models through normal requests—as illustrated in Figures 1 and 3. **Our work contributes to the safety community by addressing this critical alignment vulnerability.**
> >
> >   Additionally, we have tested more recent models and found that they remain far from safe to serve as agents.
> >
> >   |                        | Avg Score (↓) | Avg Full Score (↓) | Avg Refusals (↑) |
> >   | :--------------------: | :-----------: | :----------------: | :--------------: |
> >   |  Qwen-3-30B-A3B-2507   |     39.8      |        21.0        |       50.6       |
> >   |       GPT-5-mini       |     23.8      |        11.4        |       54.0       |
> >   | Deepseek-V3.1-Terminus |     36.8      |        25.0        |       56.8       |
> >   | Qwen-2.5-7B-AgentAlign |      6.7      |        1.7         |       85.8       |
> >
> >   This further underscores the timeliness and necessity of our work.
> >
> > > Could you please just add this Limitations section in the rebuttal version?
> >
> > - We have uploaded a revised version that includes a Limitations section discussing adversarial robustness and the multi-turn interaction and decomposition attacks you mentioned. According to the Author Policy, only the Ethics Statement and Reproducibility Statement are recommended to be placed in the main text after the References. Therefore, we have included the Limitations section in Appendix G. If we are able to secure an additional page in the camera-ready version, we will move it into the main text.
> >
> >
> >
> > Thank you  again for helping us improve this paper.

---

### Official Review · Reviewer_eCxY · 2025-11-01

**Soundness:** 2
**Presentation:** 3
**Contribution:** 4
**Rating:** 6
**Confidence:** 3

**Summary:**

This paper proposes AgentAlign, a framework for improving the safety alignment of agentic large language models (LLMs) that can execute multi-step actions and use external tools. The key idea is to represent potential harmful or benign behaviors as Abstract Behavior Chains. The authors construct a simulation environment with 86 functional APIs across 9 tool categories to safely generate and validate a large-scale dataset (~18K samples) of harmful and benign agentic instructions. AgentAlign demonstrates significant improvements on the AgentHarm and ToolSword benchmarks. Specifically, the method substantially increases the refusal rate on harmful tasks while maintaining utility on benign tasks.

**Strengths:**

* Clear motivation and presentation: The paper provides a well-motivated discussion of the emerging safety challenges in agentic LLMs, supported by concrete examples and quantitative evidence. The writing is clear and easy to follow.
* Originality: The idea of modeling safety through Abstract Behavior Chains is novel and insightful, as it captures multi-step harmful behaviors at the behavioral logic level rather than relying on surface text filters.
* The proposed simulation environment and accompanying dataset are strong contributions, enabling safe and systematic synthesis of agentic tasks for alignment training. The semantic and execution validation framework is particularly rigorous and enhances data reliability.
* The experiments are comprehensive, covering multiple open-source models and benchmarks (AgentHarm, ToolSword).

**Weaknesses:**

* While the paper is strong overall, it would benefit from a more comprehensive discussion of related work on plug-and-use safety guardrails for agents, such as GuardAgent, Conseca, and Agrail, to better position AgentAlign within this growing research space.
* The training setup is not clearly described in the main text; readers may find it difficult to understand how the proposed dataset and objectives are applied during fine-tuning. Including a concise summary of the training process (currently only in the appendix) would significantly improve clarity.
* Similarly, the simulation environment is central to the paper’s contribution, but its implementation details and accessibility are limited. Open-sourcing or providing more technical documentation on the environment and dataset would enhance reproducibility and impact.
* On the empirical side, there is a slight drop in benign task performance for some models after applying AgentAlign, and results on Ministral and Qwen remain below Claude-3.5-Haiku. Moreover, it would strengthen the work to include comparisons against other representative guardrail systems such as Llama-Guard 3.

**Questions:**

Most of my questions overlap with the weaknesses mentioned above. In addition, I have one question regarding the data generation process:
* Could the authors clarify why Claude-3.5-Haiku was chosen to generate refusal responses for harmful instructions, while Mistral-Large was used to generate benign trajectories?

---

> ### Author Response · Authors · 2025-11-22
>
> Thank you for your careful assessment of our work and the feedback you've provided. We appreciate your thoughtful review. Below are our responses to the questions you raised, which we hope will address your concerns.
>
> ### Response to weakness
>
> > While the paper is strong overall, it would benefit from a more comprehensive discussion of related work on plug-and-use safety guardrails for agents, such as GuardAgent, Conseca, and Agrail, to better position AgentAlign within this growing research space.
>
> - Thank you for this suggestion. We will expand the related work section in the updated version to include a more comprehensive discussion of guardrails for agents. This will help better position AgentAlign within this growing research area and provide clearer context for our contributions.
>
> > The training setup is not clearly described in the main text; readers may find it difficult to understand how the proposed dataset and objectives are applied during fine-tuning. Including a concise summary of the training process (currently only in the appendix) would significantly improve clarity.
>
> - Due to space constraints, we placed the detailed training details in Appendix D. We appreciate you highlighting this. If we are able to secure an additional page in the camera-ready version, we will add a concise summary of the key training details in the main text to improve readability.
>
> > Similarly, the simulation environment is central to the paper’s contribution, but its implementation details and accessibility are limited. Open-sourcing or providing more technical documentation on the environment and dataset would enhance reproducibility and impact.
>
> - Yes, we are committed to full reproducibility. Upon publication, we will release all prompts, code, simulation environments, and datasets as open-source resources to facilitate future research in this direction.
>
> > On the empirical side, there is a slight drop in benign task performance for some models after applying AgentAlign, and results on Ministral and Qwen remain below Claude-3.5-Haiku. Moreover, it would strengthen the work to include comparisons against other representative guardrail systems such as Llama-Guard 3.
>
> - We have conducted additional experiments comparing our AgentAlign-aligned Llama-3-8B model (Functionary-8B-aligned) with Llama-Guard-3-8B. The comparative results on AgentHarm are shown below:
>
>   |                        | Harmful Requests Refusal (%) | Benign Requests Refusal (%) |
>   | :--------------------: | :--------------------------: | :-------------------------: |
>   |    Llama-Guard-3-8B    |             97.7             |            24.4             |
>   | Functionary-8B-aligned |             88.6             |            12.5             |
>
> ​	As the results demonstrate, Llama-Guard-3-8B suffers from significant overfitting to harmful content, leading to a 	high false positive rate of 24% on benign user requests—a critical limitation that severely impacts practical usability. 	Our method, by contrast, successfully internalizes safety awareness while maintaining superior utility, achieving a 	more favorable safety-utility balance. Furthermore, our approach enables direct deployment without the architectural 	overhead of an additional external guardrail module, providing a more efficient and self-contained solution.
>
>
> ### Response to questions
>
> > Could the authors clarify why Claude-3.5-Haiku was chosen to generate refusal responses for harmful instructions, while Mistral-Large was used to generate benign trajectories?
>
> - We selected Claude-3.5-Haiku and Mistral-Large based on a careful consideration of both performance and cost factors. Specifically, Claude-3.5-Haiku demonstrated the best safety performance among available models while maintaining reasonable API costs, making it an ideal choice for our safety-focused response. Mistral-Large, on the other hand, provides generous free research credits while offering strong agentic performance, which was crucial for our experiments. Given the scale of our data collection—involving multi-step trajectories with long contexts—budget considerations were indeed an important practical constraint that we had to address.

---

### Official Review · Reviewer_BDDN · 2025-11-01

**Soundness:** 2
**Presentation:** 3
**Contribution:** 2
**Rating:** 2
**Confidence:** 5

**Summary:**

The authors introduce AgentAlign, a framework that models malicious agent behaviors through "abstract behavior chains" - structured representations of multi-step harmful action sequences. These chains are instantiated in simulated environments to generate training data that balances safety and utility. Experiments across three model families show substantial improvements while maintaining performance on benign tasks.

**Strengths:**

1. Relevant problem: The safety gap between conversational and agentic LLMs is a real concern worth investigating.
2. Systematic data generation: The abstract behavior chain framework provides a structured approach to generating multi-step harmful scenarios, which is important in AI safety research.

**Weaknesses:**

1. The method is missing critical comparisons. This is fundamentally a data generation method, yet there are no comparisons to:
- Existing safety datasets: GuardSet-X [1], ToolAlign (only briefly mentioned in the related work), and other multi-step safety datasets. How does training on your dataset compare to training on these?
- Guardrail systems: Why not compare against ShieldAgent [2], LlamaGuard, or other input filtering approaches? These operate at inference time without requiring model retraining. The paper doesn't justify why fine-tuning is necessary when you could simply filter inputs with an existing safety classifier.
- Other data generation approaches: What about simple augmentation of existing red-teaming datasets? Or using LLMs to generate harmful agent scenarios with different prompting strategies?

2. The evaluated models (GPT-4o, Qwen-2.5) are already outdated. More recent models should be evaluated.

3. Transferability issue not addressed. If the agent is equipped with new sets of tools (or APIs), will the model still show a good refusal rate?

[1] Kang, Mintong, et al. "Guardset-x: Massive multi-domain safety policy-grounded guardrail dataset." arXiv preprint arXiv:2506.19054 (2025).
[2]

**Questions:**

1. Why not compare to LlamaGuard or ShieldAgent as input filters? This seems like the most obvious baseline.
2. How does performance compare when training on other existing safety datasets?
3. What happens with completely different tool ecosystems? If I deploy your trained model with an entirely new set of APIs, does the safety transfer?
4. Can you show this works on current frontier models? The models tested are outdated.

---

> ### Author Response · Authors · 2025-11-22
> **Response to Weakness**
>
> Thank you for your careful assessment of our work and the feedback you've provided. We appreciate your thoughtful review. Below are our responses to the questions you raised, which we hope will address your concerns.
>
> > The method is missing critical comparisons. This is fundamentally a data generation method, yet there are no comparisons to: Existing safety datasets: GuardSet-X [1], ToolAlign (only briefly mentioned in the related work), and other multi-step safety datasets. How does training on your dataset compare to training on these?
>
> - The reviewer mentioned GuardSet-X, which is a dataset for training guardrail models. We discuss this in detail when addressing input filtering approaches in later response. As for ToolAlign, we provided a comparison highlighting the key differences in Appendix E and Figure 10.
>
> > Guardrail systems: Why not compare against ShieldAgent [2], LlamaGuard, or other input filtering approaches? These operate at inference time without requiring model retraining. The paper doesn't justify why fine-tuning is necessary when you could simply filter inputs with an existing safety classifier.
>
> - We would like to clarify the fundamental distinction between our approach and guardrail-based methods. Our core objective is to teach models **intrinsic** safety capabilities. More specifically, when deploying or releasing a model, **we aim to ensure it is inherently safe in agentic scenarios, rather than requiring an external input filter to function safely**. In this sense, our work follows a fundamentally different research direction from guardrail-based approaches. Nevertheless, to provide a complete picture, we have included a comparison with Llama-Guard-3-8B, with results shown below:
>
>   |                        | Harmful Requests Refusal (%) | Benign Requests Refusal (%) |
>   | :--------------------: | :--------------------------: | :-------------------------: |
>   |    Llama-Guard-3-8B    |             97.7             |            24.4             |
>   | Functionary-8B-aligned |             88.6             |            12.5             |
>
>   As the results demonstrate, Llama-Guard-3-8B suffers from significant overfitting to harmful content, leading to a high false positive rate of 24% on benign user requests—a critical limitation that severely impacts practical usability. Our method, by contrast, successfully internalizes safety awareness while maintaining superior utility, achieving a more favorable safety-utility balance. Furthermore, **our approach enables direct deployment without the architectural overhead of an additional external guardrail module, providing a more efficient and self-contained solution**.
>
> > Other data generation approaches: What about simple augmentation of existing red-teaming datasets? Or using LLMs to generate harmful agent scenarios with different prompting strategies?
>
> - AgentAlign is the result of continuous refinement. Simply modifying existing red-teaming datasets can easily lead to overfitting on harmful prompts like Llama-Guard, which damages the model's utility. One of our main contributions is using abstract behavior chains with two types of demonstrations (benign and harmful) to effectively learn the boundary between helpfulness and harmlessness. If you carefully examine our main results (Table 1), our method significantly improves the performance of Qwen-2.5-7B and Functionary-Small-v3.2 (Llama-3.1-8B) on **benign** tasks (53.4% → 64.2%, 45.9% → 53.5%). **This shows that our approach is also an important resource for improving general agentic capabilities.**
>
> > The evaluated models (GPT-4o, Qwen-2.5) are already outdated. More recent models should be evaluated.
>
> - The models used in our work—Qwen-2.5, Llama-3 (Functionary), and Mistral—were all released in 2024 and remain the most widely used and broadly adopted open-source models to date. Therefore, this should not be considered a weakness.
>
> > Transferability issue not addressed. If the agent is equipped with new sets of tools (or APIs), will the model still show a good refusal rate?
>
> - In fact, the tools in our synthesized simulation environments are different from those in the downstream benchmarks we use for testing. This means that **when evaluating on AgentHarm and ToolSword, the models are already equipped with new sets of tools**, which demonstrates that our method has strong generalization capabilities.

---

> ### Author Response · Authors · 2025-11-22
> **Response to Questions**
>
> > Why not compare to LlamaGuard or ShieldAgent as input filters? This seems like the most obvious baseline.
>
> - We have added a comparison with input filter methods (Llama-Guard) in our Response to Weakness.
>
> > How does performance compare when training on other existing safety datasets?
>
> - We have provided our explanation in Response to Weakness.
>
> > What happens with completely different tool ecosystems? If I deploy your trained model with an entirely new set of APIs, does the safety transfer?
>
> - Similarly, please see our Response to Weakness.
>
> > Can you show this works on current frontier models? The models tested are outdated.
>
> - Following your suggestion, we have tested the safety performance of more recent models. The results on Harmful Tasks from AgentHarm are shown below:
>
>   |                        | Avg Score (↓) | Avg Full Score (↓) | Avg Refusals (↑) |
>   | :--------------------: | :-----------: | :----------------: | :--------------: |
>   |  Qwen-3-30B-A3B-2507   |     39.8      |        21.0        |       50.6       |
>   |       GPT-5-mini       |     23.8      |        11.4        |       54.0       |
>   | Deepseek-V3.1-Terminus |     36.8      |        25.0        |       56.8       |
>   | Qwen-2.5-7B-AgentAlign |      6.7      |        1.7         |       85.8       |
>
>   As shown in the results, **even these latest models respond to nearly half of the harmful requests, which remains far from safe to serve      as an agent. This further underscores the timeliness and necessity of our work**.

---

> > ### Comment · Reviewer_BDDN · 2025-11-26
> >
> > I would like to thank the authors for their detailed response, especially the results of the new models. Some of my concerns are addressed. Given that the synthesized data could be valuable to the research community, I will raise my rating to 4 (weak reject).
> >
> > However, I am not convinced by the claim that the finetuning approach achieves a better safety-utility trade-off than the guardrail models like LlamaGuard. Generally, having a model post-trained for a specific task (like guardrail models) would achieve better performance. If the authors would like to claim it improves the model's intrinsic safety awareness without sacrificing utility, then different benign benchmarks should be tested (i.e., MMLU, AIME, other agentic AI benchmarks, etc.), which is costly, and I believe the performance will degrade a lot. Even on the current benchmark, it's not convincing that the model is better than LlamaGuard according to the results.
> >
> > With that being said, I think one possible direction could be fine-tuning the guardrails with your data and demonstrating the performance improvement, then the setting would be more reasonable to me. If that's the case, I would be happy to further raise my rating.

---

> ### Author Response · Authors · 2025-12-02
>
> Thank you for raising the score. Below are our responses to your remaining concerns:
>
> > If the authors would like to claim it improves the model's intrinsic safety awareness without sacrificing utility, then different benign benchmarks should be tested (i.e., MMLU, AIME, other agentic AI benchmarks, etc.), which is costly, and I believe the performance will degrade a lot.
>
> - Following your suggestion, we have added an evaluation on another general agentic benchmark, the [Berkeley Function-Calling Leaderboard](https://gorilla.cs.berkeley.edu/leaderboard.html). The results are as follows:
>
>   |                        | Single-turn (simple) | Single-turn (multiple) | Multi-turn_base |
>   | :--------------------: | :------------------: | :--------------------: | :-------------: |
>   |     Functionary-8B     |        70.7%         |         95.0%          |      15.0%      |
>   | Functionary-8B-aligned |        74.5%         |         95.5%          |      29.0%      |
>
>   *Note: "simple" represents the average score across three subsets: simple_python, simple_java, and simple_javascript.
>
>   As demonstrated by the results, **training with AgentAlign does not compromise the model's general agentic (or tool-calling) capabilities**. Meanwhile, benefiting from the enhancement of our abstract behavior chains, the model shows significant improvement on multi-turn tasks (15% → 29%), which further validates that AgentAlign improves the model's intrinsic safety awareness without sacrificing utility.
>
> > With that being said, I think one possible direction could be fine-tuning the guardrails with your data and demonstrating the performance improvement, then the setting would be more reasonable to me.
>
> - We respectfully note that if you believe that fine-tuning LlamaGuard with our data would indeed be effective, this precisely validates our contribution—synthesizing highly realistic synthetic tasks through abstract behavior chains and simulated environments is among our core contributions. More importantly, **our method offers a fundamental advantage over guardrails: it eliminates the need for additional deployment and inference overhead**. We can directly release the aligned model, whereas guardrail-based approaches require developers to integrate external components. This distinction becomes increasingly critical in the prevalent model-as-a-service paradigm.
>
> We hope our responses have addressed your concerns. Thank you again for your feedback.

---

### Author Response · Authors · 2025-11-22

We sincerely thank all reviewers for their valuable time and thoughtful feedback, especially given the significantly increased submission volume this year. Below, we address several common themes that emerged across the reviews:

- **Regarding guardrail models:** All three reviewers inquired about the role of guardrail-based approaches in this context. We would like to clarify that **our primary objective is to enhance the intrinsic safety capabilities of agentic models, rather than training external guardrails.** The unique challenges we address include ensuring the authenticity of task and environment synthesis, and achieving a better balance between helpfulness and harmlessness in agentic settings. Nevertheless, we have included comparisons with guardrail models (e.g., Llama-Guard-3) in our individual responses, and will expand our related work section to provide a more comprehensive discussion of this landscape as suggested.
- **On the scope of our work:** Our work represents an early and systematic effort to articulate the safety alignment challenges specific to agentic systems and propose corresponding solutions. The components we developed—including **task synthesis**, **simulation environment construction**, **human validation**, and **evaluation** across multiple benchmarks—represent a substantial research effort. We believe these resources will serve as valuable foundations for the community's continued research on language agent alignment. That said, we acknowledge that no single work can comprehensively address all aspects of this complex problem. For instance, adversarial robustness against sophisticated jailbreaking attacks, as raised by Reviewer fXJL, is an important direction that we will discuss candidly in a new Limitations section.
- **Additional evidence of urgency:** In our response to Reviewer BDDN, we have included evaluation results for more recent models, which demonstrate that they remain far from safe for agentic deployment.  Furthermore, [recent reports from Anthropic](https://www.anthropic.com/news/disrupting-AI-espionage) documenting large-scale cyberattacks powered by AI agents (corresponding to the motivation illustrated in our Figure 1) further underscore the pressing need for agent safety alignment.

**In summary, we believe our work takes a solid step toward bridging the safety alignment gap between conversational and agentic LLMs.** We would be deeply grateful if the reviewers would recommend our work to the broader community.

---

### Meta-Review · Area_Chair_Jxtu · 2026-01-07

**Summary:**

This paper proposes AgentAlign, a new framework for aligning large language models with agentic capabilities by modeling and simulating abstract behavior chains representing harmful action sequences. In experiments, AgentAlign improves model safety on malicious requests while preserving utility on benign tasks.

Two reviewers raise serious concerns that the paper includes only weak baselines, omitting evaluation and comparisons with other obvious strong baselines. The rebuttal, however, is not sufficiently convincing. Two reviewers who are negative remain negative about the paper. Given that the major concerns raised in the initial reviews have not been properly addressed, the AC recommends to reject.

**Reviewer Concerns:**

Reviewer BDDN: missing critical comparisons
- GuardSet-X, ToolAlign , ShieldAgent, LlamaGuard

The rebuttal clarifies that the paper's core goal is "intrinsic" safety capabilities.
"Our core objective is to teach models intrinsic safety capabilities. More specifically, when deploying or releasing a model, we aim to ensure it is inherently safe in agentic scenarios, rather than requiring an external input filter to function safely." and therefore different from guardrail-based approaches.

However, this explaination does not convince Reviewer BDDN, as the reviewer states "However, I am not convinced by the claim that the finetuning approach achieves a better safety-utility trade-off than the guardrail models like LlamaGuard." "Even on the current benchmark, it's not convincing that the model is better than LlamaGuard according to the results."

Reviewer fXJL: "Only compared to weak prompting baselines. For example, I'd appreciate it if you could at least add one common guardrail baseline, such as Llama Guard 4?"

The rebuttal did provide the results on Llama-Guard-3-8B. However, while the results show Llama-Guard-3-8B has higher refusal rate of 24%, it also shows higher Harmful Requests Refusal rate. It is unclear if the proposed alignment actually improves upon existing guardrail-based methods, or it's just a different trade-off.

After the discussions, Reviewer fXJL does not raise the score.
"Overall, given the concerns outlined above, I'm not comfortable raising my score."

**Reviewer Scores:**

Reviewer BDDN: 2: reject, not good enough -> 4 (weak reject)
"Given that the synthesized data could be valuable to the research community, I will raise my rating to 4 (weak reject)."

Reviewer eCxY: 6: marginally above the acceptance threshold.

Reviewer fXJL: 4: marginally below the acceptance threshold.
"Overall, given the concerns outlined above, I'm not comfortable raising my score."

The two negative reviewers remain negative after the rebuttal.

---

### Decision · Program_Chairs · 2026-01-26

Reject